# Can Donkey Behavior and Cognition Be Used to Trace Back, Explain, or Forecast Moon Cycle and Weather Events?

**DOI:** 10.3390/ani8110215

**Published:** 2018-11-19

**Authors:** Francisco Javier Navas González, Jordi Jordana Vidal, Gabriela Pizarro Inostroza, Ander Arando Arbulu, Juan Vicente Delgado Bermejo

**Affiliations:** 1Department of Genetics, Faculty of Veterinary Sciences, University of Córdoba, 14071 Córdoba, Spain; kalufour@yahoo.es (G.P.I.); anderarando@hotmail.com (A.A.A.); juanviagr218@gmail.com (J.V.D.B.); 2The Worldwide Donkey Breeds Project, Faculty of Veterinary Sciences, University of Córdoba, 14071 Córdoba, Spain; worldwidedonkeybreedsproject@hotmail.com; 3Departament de Ciència Animal i dels Aliments, Facultat de Veterinària, Universitat Autònoma de Barcelona, 08193 Bellaterra, Spain; jordi.jordana@uab.cat

**Keywords:** cognition, cold wave, learning abilities, lunar phases, meteorological conditions

## Abstract

**Simple Summary:**

Donkeys have been traditionally attributed the ability to inform humans about the environment. Carefully observing the behavior and cognitive reactions of donkeys in their habitat may enable to quantify such reactions to develop informative mathematical models. These models can be used to explain present environmental situations, trace back past events or even predict future conditions. Our results suggest, environmental stressing situations may affect donkeys in a way that they register the cognitive adaptations or sequels derived from such situations. Furthermore, such environmental events may not only affect the present cognitive status of the animals, but they may drive this cognitive record affecting the behavioral patterns donkeys display through their lives. Our model is able to explain 75.9% of the variability in response type and intensity, mood, or learning capabilities. Conclusively, donkeys can be used as an environment informative sensitive tool and may therefore, predict and register slight human-unappreciable climatic variations to which they may behaviorally adapt beforehand.

**Abstract:**

Donkeys have been reported to be highly sensitive to environmental changes. Their 8900–8400-year-old evolution process made them interact with diverse environmental situations that were very distant from their harsh origins. These changing situations not only affect donkeys’ short-term behavior but may also determine their long-term cognitive skills from birth. Thus, animal behavior becomes a useful tool to obtain past, present or predict information from the environmental situation of a particular area. We performed an operant conditioning test on 300 donkeys to assess their response type, mood, response intensity, and learning capabilities, while we simultaneously registered 14 categorical environmental factors. We quantified the effect power of such environmental factors on donkey behavior and cognition. We used principal component analysis (CATPCA) to reduce the number of factors affecting each behavioral variable and built categorical regression (CATREG) equations to model for the effects of potential factor combinations. Effect power ranged from 7.9% for the birth season on learning (*p* < 0.05) to 38.8% for birth moon phase on mood (*p* < 0.001). CATPCA suggests the percentage of variance explained by a four-dimension-model (comprising the dimensions of response type, mood, response intensity and learning capabilities), is 75.9%. CATREG suggests environmental predictors explain 28.8% of the variability of response type, 37.0% of mood, and 37.5% of response intensity, and learning capabilities.

## 1. Introduction

The hypothetical conditioning effects of weather, moon and climate oscillations on animal behavior and cognition have been widely but unscientifically reported. Popular knowledge has even provided untested testimony of the possibility to predict short-term future meteorological conditions basing on how animals react to the environment around them. This framework has promoted the appearance of the first empirical studies on the clinical and productive implications of such environmental factors in different animal species.

Great scale migration of animal populations, adaptation, or even census reduction or extinction have become proved symptoms of how life cycles may be affected by this progressively changing environmental situation. However, the alteration of the particular environmental characteristics of specific areas has also been suggested to lead the lower scale evolutionary process of local animal life cycles [1].

Research has focused on the study of the climatological alteration of physiological processes such as reproduction, and animal biorhythms in populations of different species [2]. By contrast, cognitive or behavioral alterations affecting animal populations may remain unnoticed due to being attributed to other more probable causes.

The study of the effects of factors such as season and weather on animal behavior and mood has typically focused on understanding the changes in the ethological patterns conditioning animal routine and daily activities. These changes may globally appear as a consequence of the evolution of certain areas, which may no longer fulfil the unique set of requirements of the animal populations inhabiting them [3].

Parallel to these more or less quantifiable effects, there is also a simultaneous repercussion on animal cognitive or behavioral health [4]. These effects may not only alter the components of disorder incidence but may also condition animal physiology, as they increase the levels of sensitivity or even distort the cognitive status of specific populations producing long-lasting consequences.

When we consider these behavioral and cognitive registers under a local specific context, we can trace back their origin up to potential weather or meteorological condition related situation or event [5].

Scientists have paid attention to the study of the environmental changes that may distort seasonal and circadian rhythms in different species. However, the effects of factors such as the moon cycle on animal behavior have only been approached assessing the alterations occurring on daily animal patterns or physiological rhythms [6]. Not to mention the inexistence of research assessing other traditionally folklore-reported environmental effects on cognition, such as the hypersensibility to anticipate particular events. The role on neuroanatomy, ethology, and endocrinology and the activity and effects of neurohormones releasing cycles may be triggered and regulated by the electromagnetic radiation and the gravitational pull of the moon and light cycles during the different moon phases, which may reflect in psychological processes such as mood or cognitive abilities.

The first aim of this research is to study at which level environmental factors such as season, year, moon cycle, meteorological factors, and climate oscillations may affect the response type and intensity, mood and learning abilities of donkeys. Second, we used categorical principal component analyses (CATPCA) to study the possibility to reduce our set of environmental variables to a smaller set that still contains most of the information in the previous one, hence reducing the likelihood of Type I error that can derive when testing for the effects of a large number of explanatory and predictor variables. Third, using this reduced information, we designed regression equations using categorical regression (CATREG) to explain, trace back, and predict the possible behavioral repercussions that certain environmental situations may have, and how these consequences may alter the behavioral patterns that donkeys display through their lives, in order to provide clues on how behavior can become a useful tool for daily care.

## 2. Materials and Methods

### 2.1. Animal Sample

Our study sample comprised 78 Andalusian uncastrated jacks and 222 unneutered jennies (*n* = 300), born from 1990 to 2012 and officially registered in the national studbook of the Andalusian donkey breed. As the age range was not normally distributed (*p* < 0.05, Kolmogorov–Smirnov test for normality) we used minimum, Q1, median, Q3 and maximum to describe the age range in our sample. Minimum age in the range was 0.27 months, Q1 age was 29.76 months, median age was 77.04 months, Q3 age was 129.07 months, and the maximum age was 270.40 months.

### 2.2. Information Registration

We registered the information on the response type and response intensity, mood/emotional collateral responses and learning ability from the donkeys in our sample during the development of a six-stage operant conditioning test (Table 1). Reinforcement treatments, stimuli descriptions, their classification, and their constituting elements are provided in Table 1 and Table 2. The same trained judge registered all the information concerning the four behavioral variables and 15 noncognitive factor for all the stages and animals. The donkeys were each given a maximum of 450 s to complete the operant conditioning test (75 s per stage and treatment implemented). No additional time was provided for the donkeys to complete the test. The information registered corresponded to the first immediate reaction described by each animal when each of the stages was started. In 75 s, an animal can shift attention many times. However, to simplify the observations, our study tested for the first reaction of the animals, further reactions shown through the development of the test were discarded.

The records for each animal consisted of information on 18 categorical variables divided into two sets. The first set of 4 dependent behavioral categorical variables assessed the cognitive performance of donkeys through their response type, response intensity, mood/emotion, and learning ability. The variables in this first set could be conditioned by a second set of independent variables comprising 14 environmental factors. A summary of the variables and categories included in the first variable set is described in Appendix A, while Appendix A shows a summary of the factors and categories included in the second categorical factor set. Appendix A shows the descriptive statistics, and numerical parametrization of all the variables analyzed. Appendix A presents Category description and definition for response type, the intensity of response, mood/emotion, and learning variables directly controlled during the operant conditioning test.

### 2.3. Categorical Behavioral Variables

The reaction developed by the donkeys when they faced the six consecutive treatments provided information on four categorical behavioral variables (Appendix A). To name the mood/emotion variable, we considered the definitions by Cabanac [7] and Mendl et al. [8]. Appendix A shows a description of the scales used to score the response type and mood/emotion variables. The intensity of response and learning ability variables were subdivided into five categories each described as shown in Appendix A. The appraiser scored the animals relying on the intensity of their responses from low intensity responses to high intensity responses whatever the mood/emotion displayed by them was (Appendix A). As animals were only scored once, opposite behaviors were not scored correlatively in the same animal. That is to say, the response of an animal displaying a high intensity calm mood/emotion (very calm animal) was not registered as a low intensity nervous mood/emotion (slightly nervous mood/emotion) simultaneously. The reason for this is the fact that an animal cannot be nervous and calm at the same time whatever it is the intensity level at which such animal expresses its mood/emotion status (see Appendix A).

### 2.4. Qualitative Behavioral Assessment

The same trained judge registered each donkey’s mood/emotion following the protocols developed by Navas et al. [9] which based on Minero et al. [10]. Navas et al. [9] generated the descriptor lists for the use in subsequent studies as the present one. Appendix A shows a summary of the mood/emotion descriptors used concerning Table 2.

### 2.5. Noncognitive Categorical Factors

Environmental categorical factors could be divided into two groups. Meteorological and environmental conditions included year of evaluation, the season of evaluation, weather conditions, temperature, moon phase at evaluation, relative humidity, windspeed, sunlight hours, barometric pressure, rainfall on the day of evaluation, and rainfall on the following day. Animal birth characteristics included season of birth, year of birth and moon phase at birth. Appendix A shows the categories for independent noncognitive factors in the second set.

The information was registered during the yearly behavior assessment sessions carried out on four random days per year, from June to November for three consecutive years from 2013 to 2015 at twenty-two different farms all over Andalusia (Southern Spain).

The 22 farms involved, reared their animals under four husbandry systems (extensive, semi extensive, semi intensive and intensive) and were located in 5 Andalusian provinces (Southern Spain). The 6% of the donkeys were tested during the breed’s Official Morphological Contest held by the Union of Andalusian Donkey Breeders (UGRA).

### 2.6. Meteorological and Moon Cycle Records

Day records for temperature, relative humidity, windspeed, sunlight hours, barometric pressure, rainfall per day and rainfall prediction (on the following day) were obtained from the State Meteorological Agency (AEMET) (http://www.aemet.es/). Moon phase at evaluation and moon phase at birth records were obtained from the Astronomical Applications Department of the US Naval Observatory (http://aa.usno.navy.mil).

### 2.7. Operant Conditioning Behavioral Test

The operant conditioning behavioral test was carried out in an open area to which the donkeys were previously accustomed (it was part of the area over which the donkeys developed their daily activities). During the operant conditioning test, the donkeys were made cross over a 200 × 200 cm oilcloth with a wooden print on it using increasingly aversive reinforcement methods (from stimuli 1 to 6). We exposed each animal to six reinforcement treatments consecutively, one at each of the six stages within the operant conditioning test. At each stage, handler A and handler B used each of the six different reinforcement treatments to lead the donkeys to cross over an oilcloth laying on the floor. These treatments/stimuli could comprise unknown elements (the animal had not been familiarized to them) or known elements (to which the animal had already been familiarized). These elements could be visual (elements fell within the visual areas of the donkeys) and/or acoustic (elements generated sounds, i.e., “motivator” or claps, although they may or may not fall within visual areas) and were presented to the donkeys from different positions (from the front or from a rear position always at 2 m away from the animals). A cameraman (Handler C) simultaneously videotaped the experiences (1080 p, 50 Hz, shutter speed: 1/250 s) to assess the donkey’s performance after the field experiences and to test for intra-observer discrepancies. Cameraman (Handler C) controlled timing. A detailed description of the operant conditioning test, the reinforcement treatments, stimuli descriptions and classification and their constituting elements are described in Navas et al. [9] and Navas González et al. [11], and summarized in Table 1 and Table 2.

### 2.8. Test and Scoring System Reliability

Statistical tests did not report intra-observer discrepancies as all the scores obtained on the field matched those obtained after reviewing the tapes again. Aiming at eliminating the effect of appraiser to reduce the likelihood of subjective evaluations, 50 individuals (16.67% of the total sample) were tested using the operant conditioning test described scoring for the categorical variables of response type, mood and response intensity at a preliminary stage of the study. Cohen’s κ determined whether the repeatability of the model was enough to delete the effect of the appraiser from the model, providing a measure of the accuracy of scoring of the appraisers. Then 95% confidence intervals (95% kappa IC) were computed according to 95% kappa IC = κ ± 1.96 SEκ, where; SEκ = [(po(1 − po)/*n*(1 − pe)^2^]^0.5^ with the Crosstabs procedure of SPSS Statistics for Windows, Version 24.0, IBM Corp. (2016, Armonk, NY, USA). This preliminary analysis aimed at testing for interobserver reliability, i.e., the reliability of the scoring system, which proved to be highly reliable as there was highly statistically significant perfect agreement between the three appraisers’ judgements when scoring for response type and response intensity for the six stimuli/treatments presented. Each stimulus corresponded to one of the six stages in the test (Table 1). When testing for mood/emotion, there was highly statistically significant almost perfect agreement among the three observers at the preliminary test for repeatability for all the traits and stimuli, except when testing for mood at the presentation of stimulus/treatment 3. In this case, the strength of agreement between appraisers 1 and 2 and 2 and 3 was substantial and at the presentation of stimuli/treatments 1 and 6, for appraisers 2 and 3 between whom inter-observer agreement was substantial. The slight distortion occurring may be attributed to the change in the kind of reinforcement applied to make the donkeys cross over the oilcloth on the floor occurring in stimuli/treatments 1, 3, and 6. At the presentation of stimulus/treatment 1, the animal passed from being at rest to start the operant conditioning test. At the presentation of stimulus/treatment 3, the animals went from being exposed to negative reinforcement (stimulus/treatment 2) to being exposed to positive/neutral reinforcement (stimulus/treatment 3). Finally, at the presentation of stimulus/treatment 6, the stimulus changed from being presented at the visible area of the donkey to be located at a rear position (blind area). Appendix A shows the results for interobserver reliability tests at this preliminary study.

### 2.9. Statistical Analysis

Categorical variables represent a qualitative method of scoring data. As all the variables and factors considered in our study were categorical, we used nonparametric tests to assess the information recorded statistically. A Chi-square test for independence was used to analyze whether the factors in the second set (Appendix A) randomly and significantly influenced the variables in the first set (Appendix A). Chi square is neutral to the parametric or non-parametric nature of the distribution and is relatively robust to situations with a limited number of data (*n* > 50). The most appropriate statistic to use as a measure of Chi-square association is Cramér’s V. Cramér’s V is used to measure the strength of linear correlation, that is to test for the multicollinearity and significance between each variable from the first set with each variable from the second set using the Crosstabs procedure from SPSS Statistics for Windows, Version 24.0, IBM Corp. (2016, Armonk, NY, USA) according to the indications of Nolan [17]. Appendix A shows total and relative frequencies for the associations of the four dependent categorical variables with the environmental variables.

Categorical principal components analysis (CATPCA) was used to quantify categorical factors while reducing the dimensionality of the data and Categorical regression to establish the most important descriptive and discriminative noncognitive factors on the variables considered using the Optimal Scaling procedure from the Dimension reduction task from SPSS Statistics for Windows, Version 24.0, IBM Corp. (2016, Armonk, NY, USA). Reducing the dimensionality of relatively large sets of variables prevents type I errors from occurring, as we may strip our model to the core independent variables affecting the dependent variables studied by our model. A lower number of variables means we may need stronger evidence against the null hypothesis H_0_ (via a lower *p*-value) before we will reject the null. Therefore, if the null hypothesis is true, we will be less likely to reject it by chance. This reduced information was used later at the categorical regression (CATREG) analysis.

We used CATREG to describe regression models to study how the variables assessed depended on the factors considered. The resulting regression equations could be used to trace back, explain, or predict behavior or cognitive abilities for any combination of the 14 independent factors. Categorical regression was carried out using the Optimal Scaling procedure from the Regression task from SPSS Statistics for Windows, Version 24.0, IBM Corp. (2016, Armonk, NY, USA).

### 2.10. Justification for Statistical Tests

The most appropriate statistic to use as a measure of Chi-square association is Cramér’s V. Cramer’s V is a measure of association for nominal variables. Effectively it is the Pearson chi-square statistic rescaled to have values between 0 and 1 as follows:(1)$$V=\sqrt{\frac{\chi^{2}}{n_{obs(\min(n_{cols},\;n_{rows}))}-1}}$$
where χ^2^ is the Pearson chi-square, *n_obs_* represents the number of observations included in the table, and where *n_cols_* and *n_rows_* are the number of columns and rows in the table, respectively. For a 2 by 2 table, of course, this is just the square root of chi-square divided by the number of observations, which is also known as the phi coefficient. Cramer’s V squared is the average of the squares of the canonical correlation coefficient between two categorical variables. Such canonical-correlation analysis will find the strength that linear combinations of the X_i_ and Y_j_ have on each other. When using Cramér’s V small effect associations range from 0.0 to 0.10, medium effect associations from 0.3 to 0.5 and large effect associations from 0.5 to anything above. The same author would recommend that the interpretation of effect size should consider a statistically significant measure (*p* < 0.05) with a small effect size or higher to indicate a meaningful difference, especially for behavioral or psychological studies.

CATPCA is appropriate to reveal the inherent overlapping nature of behavioral variables, hence becomes suitable for variable selection and dimension reduction in categorical variables. This statistical test analyses the interrelationships among a large number of variables and explains these variables regarding their common underlying dimensions. The objective is to find a few linear combinations of the variables (factors) that can be used to summarize the data without losing too much information in the process. CATPCA is a nonparametric method that quantifies categorical variables through a process called optimal scaling. Optimal scaling uses category quantifications in such a way that they account for as much as possible of the variance in the quantified variables. The most relevant characteristic of CATPCA is that it can handle and discover nonlinear relationships between variables. Because CATPCA directly analyses the data matrix and not the derived correlation matrix, so that, we can avoid the usual concern to have at least five times as many observations as the variables. CATPCA suits analysis in which there are more variables than objects. In behavioral sciences many of the variables used are qualitative, nominal or ordinal, thus indicating the use of CATPCA, which has been demonstrated to be more robust than PCA when assessing categorical variables.

CATPCA eigenvalues are indicators of how many dimensions are needed. As a general rule, when all variables are either single nominal, ordinal, or numerical, the eigen value for a dimension should be larger than 1. For multiple nominal variables, there is no easy rule of thumb to determine the appropriate number of dimensions. If we replace the number of variables by the total number of categories minus the number of variables, the above rule still holds. However, this rule alone would probably allow more dimensions than are needed. When choosing the number of dimensions, the most useful guideline is to keep the number small enough so that meaningful interpretations are possible. The model summary table also shows Cronbach’s alpha (a measure of reliability), which is maximized by the procedure. In this study, the stepwise method was used to prevent the possible multicollinearity problem that could arise in the linear multiple regression model formed by transformed variables. The resulting reduced set of variables can be used to perform a categorical regression analysis to build significant behavioral descriptive equations that enable quantifying the result of the effects of specific combinations of environmental factors on behavioral variables, such as response type or intensity, mood or learning abilities.

When assessing non-parametrical data, categorical variables can be included as independent variables in a regression analysis but must be converted to quantitative data for us to be able to analyze them. Ordinary linear regression models could only be used when the dependent variable is quantitative and predictive variables are either quantitative or dummy. The analysis of such ordinary linear regression models involves minimizing the sum of squared differences between a response (dependent) variable and a weighted combination of predictors (independent). Variables are typically quantitative, with (nominal) categorical data recoded to binary or contrast variables. As a result, categorical variables serve to separate groups of cases, and the technique estimates separate sets of parameters for each group. The estimated coefficients reflect how changes in the predictors affect the response. Prediction of the response is possible for any combination of predictor values. CATREG extends the standard approach by simultaneously scaling nominal, ordinal, and numerical variables. The procedure quantifies (transforms) categorical variables so that the quantifications reflect characteristics of the original categories. The procedure treats quantified categorical variables in the same way as numerical variables. Using nonlinear transformations allow variables to be analyzed at a variety of levels to find the best-fitting model. R-squared evaluates the scatter of the data points around the fitted regression line. It is also called the coefficient of determination, or the coefficient of multiple determination for multiple regression. For the same data set, higher R-squared values represent smaller differences between the observed data and the fitted values. R-squared is the percentage of the dependent variable variation that a linear model explains. As the independent noncognitive categorical factors registered in our study were categorical and the data was sorted into categories following different criteria, we used standardized coefficients to interpret and compare their effects on our behavioral dependent categorical variables. When we apply a stepwise linear regression model to the transformed variables, the standardized and unstandardized coefficients are equal. Hence, we can interpret the unstandardized coefficients. Standardized coefficients represent regression results with standard scores. By default, most statistical software, like SPSS, automatically converts both criterion (DV) and predictors (IVs) to Z scores and calculates the regression equation to produce standardized coefficients. When most statisticians refer to standardized coefficients, they refer to the equation in which one converts both DV and IVs to Z scores. In a simple model with two factors involved the coefficients for Z scores for each variable (Z’y) may be interested as follows:

β_1_ mean a standard deviation increase in Z_X1_ is predicted to result in a β_1_ standard deviation increase in Z’y holding constant Z_X2_.

β_2_ mean a standard deviation increase in Z_X2_ is predicted to result in a β_2_ standard deviation increase in Z’y holding constant Z_X1_.

Therefore, the standardized partial coefficient represents the amount of change in Zy for a standard deviation change in Z_X_. So, if X1, one factor involved, were increased by one standard deviation, then one would anticipate a β_1_ standard deviation increase in the variable tested holding constant the effect of X2 and vice versa.

With Z_X1_ and Z_X2_, being the Z scores for each factor, and β_1_ and β_2_ the standard coefficients for each of the, respectively.

As the above example shows, conversion of raw scores to Z scores changes the unit of measure for interpretation, the change from raw score units to standard deviation units.

As a rule, we assume standardized results reported used full standardization (both DV and IVs were converted to standard scores), and that the Z formula was used for standardization. The general standardized regression equation may follow the following model Z’y = β_1_Z_X1_ + β_2_Z_X2_ + …, where Z’y is the predicted value of Y in Z scores; β_1_ represents the standardized partial regression coefficient for X1; β_2_ represents the standardized partial regression coefficient for X2; and Z_X1_ and Z_X2_ are the Z score values for the variables X1 and X2, respectively.

The intercept will always equal 0.00 when standardization is based upon Z scores, and both DV and IVs are standardized.

Once the regression equation is standardized, then the partial effect of a given X upon Y, or Z_X_ upon Zy, becomes somewhat easier to interpret because interpretation is in sd units for all predictors.

## 3. Results

### 3.1. Noncognitive Factor Analysis

Table 3 shows the results from Chi-Square and Cramér’s V, testing for the existence of linear correlations. Cramér’s V effectively measured the strength of collinearity that the noncognitive factors considered have on the behavioral variables studied, given the high significance (*p* < 0.001) that they report for all the factor-variable combinations except for season at birth and response type (Table 3). CATREG was performed to the 14 qualitative independent variables (environmental factors) with the four behavioral categorical variables (response type, mood/emotion, the intensity of response and learning ability) as dependent variables. Then stepwise linear regression to the data with the resulted quantifications was applied, and Table 4 and Table 5 present the summary results with the significant variables. Table 5 lists the standardized coefficients (β). CATREG reported all of the independent variables except for season at evaluation to be significant for response type (Appendix A). Season at evaluation and the rainfall on that day were nonsignificant for mood/emotion. Weather conditions, temperature, and barometric pressure were nonsignificant for response intensity and learning ability.

According to Cramér’s V, there was a moderate linear correlation between sunlight hours and the four behavioral variables tested (0.194 to 0.274), which was as well supported by the percentage of variance explained by this factor according to CATREG standardized coefficients. However, CATPCA addressed the correlations with three of the dimensions were inverse (from strong −0.954 to moderately weak −0.110) as reported by the values of the negative component loading (Table 3, Table 5 and Appendix A). By contrast, there was a moderate positive component, thus direct correlation with dimension 2.

For the year of birth, the Cramér’s V values ranged from 0.192 to 0.310 what reported a moderately high linear correlation. Moderately high CATREG standardized coefficients reported a moderate dependence for the four variables on this factor. Component loading for dimension 1 was negligible. However, there was a moderately strong negative loading for dimension 2 (inverse correlation) and strong positive loadings for dimensions 3 and 4 (strong direct correlation) (Table 3, Table 5 and Appendix A).

There was a moderate linear correlation between windspeed and the four behavioral variables tested (Cramér’s V ranging from 0.182 to 0.248), which was as well supported by the percentage of variance explained by this factor according to CATREG standardized coefficients. CATPCA addressed these correlations with two of the four dimensions (dimensions 1 and 3) were strongly inverse as reported by the high negative component loadings, while the other two were moderately positive thus direct (dimensions 2 and 4) (Table 3, Table 5 and Appendix A).

For the season of evaluation, the Cramér’s V values ranged from 0.196 to 0.252 what reported a moderate linear correlation. Moderate to high CATREG standardized coefficients reported a moderate to strong dependence on the four variables on this factor. Component loading for dimension 1 was high, describing a strong direct correlation. However, there was a moderately strong negative loading for dimension 3 (inverse correlation). CATPCA component loadings for dimensions 2 and 4 were positive moderately low (moderately low direct correlation) (Table 3, Table 5 and Appendix A). Season of evaluation Cramér’s values ranged from 0.049 to 0.122 (response type and mood/emotion, respectively). The CATREG standardized coefficients ranged from 0.053 to 0.075, what resembled the low to moderately low values found for Carmér’s V. CATPCA component loadings were positive and moderately low to moderate for dimensions 1, 3, and 4, and negative and moderate for dimension 2.

According to Cramér’s V, there was a moderately high linear correlation between rainfall on the following day and the four behavioral variables tested (0.183 to 0.263), which was as well supported by the percentage of variance explained by this factor according to CATREG standardized coefficients. CATPCA component loading for dimension 1 was high, describing a strong direct correlation. However, there was a moderately strong negative loading for dimension 3 (inverse correlation). Component loadings for dimensions 2 and 4 were positive moderately low (moderately low direct correlation) (Table 3, Table 5 and Appendix A).

For rainfall on the same day, the range of the linear correlations of the four variables with the factor was slightly wider (Cramér’s V from 0.177 to 0.301). This was supported by the percentage of variance explained by this factor according to CATREG standardized coefficients. CATPCA component loadings reported the same value patterns described above for rainfall on the following day (Table 3, Table 5 and Appendix A).

The range of the linear correlations of the four variables with barometric pressure ranged from 0.174 to 0.317), what was supported by the percentage of variance explained by this factor according to CATREG standardized coefficients with a dependence ranging from 0.054 to 0.365. CATPCA component loading reported positive and from moderate to strong values for the dimensions 1, 2 and 3, but the moderate negative value of the component loading for dimension 4 suggested a moderately strong negative inverse correlation (Table 3, Table 5 and Appendix A).

According to Cramér’s V, there was a moderately high linear correlation between rainfall on the following day and the four behavioral variables tested (0.183 to 0.263), which was as well supported by the percentage of variance explained by this factor according to CATREG standardized coefficients. CATPCA component loading for dimension 1 was high, describing a strong direct correlation. However, there was a moderately strong negative loading for dimension 3 (inverse correlation). Component loadings for dimensions 2 and 4 were positive moderately low (moderately low direct correlation) (Table 3, Table 5 and Appendix A).

There was a moderate linear correlation between temperature and the four behavioral variables tested (Cramér’s V ranging from 0.150 to 0.206), which was as well supported by the percentage of variance explained by this factor according to CATREG standardized coefficients. CATPCA addressed these correlations were positive and from low to high thus direct for the four dimensions (Table 3, Table 5 and Appendix A).

Year of evaluation reported Cramér’s V values ranging from 0.146 to 0.267 and CATREG standardized coefficients ranging from 0.065 to 0.242 for the behavioral variables studied (Table 3, Table 5 and Appendix A). The results of CATPCA loadings were 0.017 to 0.700 for dimensions 4 and 2, respectively. These loadings suggested a low to strong direct correlation of this factor (Table 5 and Appendix A).

The range of Cramér’s V for moon phase at evaluation for the four variables tested was narrower than the one for other factors (0.102 to 0.121). CATREG standardized coefficient range was narrow as well, ranging from 0.107 to 0.145. Values for the loadings in the CATPCA were negative and low to moderately high for dimensions 1 and 2 (weak to moderate inverse correlation), and positive and moderate to high for dimensions 3 and 4 (moderate to strong direct correlation), respectively. However, moon phase at birth reported a wider range for Cramér’s V values than other factors (from 0.111 to 0.388). By contrast, CATREG standardized coefficient range was narrow, ranging from 0.093 to 0.117. Values for the loadings in the CATPCA were positive and from low to moderate (weak to moderate direct correlation) for all the dimensions except for dimension 3, for which the value was negative and moderate (moderate inverse correlation).

Relative humidity Cramér’s V ranged from 0.117 to 0.226 for response type and mood/emotion, respectively. CATREG standardized coefficients (β) for relative humidity factor ranged from 0.106 to 0.263 for response intensity and learning, and mood/emotion, respectively. CATPCA loadings were negative and moderately high for dimensions 1 and 3 (moderately strong inverse correlation), and positive and moderate to high for dimensions 2 and 4, addressing a moderate to strong direct correlation.

For weather conditions, the range of the linear correlations of the four variables with the factor was from moderately low to moderate (Cramér’s V from 0.096 to 0.220, for response type and mood/emotion, respectively). However, the percentage of variance explained by this factor according to CATREG standardized coefficients ranged from 0.029, for response intensity and learning ability, to 0.211 for mood/emotion. CATPCA component loadings were negative and moderately low for dimensions 1 and 4 (moderate inverse correlation), and positive and moderate to high for dimensions 2 and 3 (moderate to strong direct correlation) (Table 3, Table 5 and Appendix A).

A categorical principal components analysis (CATPCA) was applied on the total data set of 14 environmental factors with the aim of establishing and interpreting the factors determining the four behavioral variables tested (response type, mood/emotion, intensity of response, and learning) to evaluate for redundancies among them. Two, three, and four-dimensional model results are shown in Table 6. Table 7 shows the factors affecting the four behavioral variables in order of importance according to the CATREG standardized coefficients (β). Since we used the stepwise method, there was no multicollinearity problem. Only 8 of the environmental factors studied contributed to the two–dimensional model in a meaningful way 11 of them meaningfully contributed to the three-dimensional model and 12 of them meaningfully contributed to the four-dimensional model (factor loadings > 0.5, Table 6), then the different components (PC1, PC2, PC3, and PC4) were best described by the factors highlighted in bold in Table 7. 

The outcomes of Cramér’s V and CATPCA analyses were used to inform the CATREG regression analyses performed and thus configure the regression equations presented in Table 8, hence the reduction of factors on each predictive equation. This reduction affects both the likelihood of Type 1 errors and the likelihood that multiple significant findings are reported as independent observations, when in fact they represent the same underlying relationship, as it was discarded in Navas et al. [9]. Table 8 presents the standardized solution for the regression equations.

The two-dimensional model has an internal consistency coefficient (Cronbach’s Alpha) of 0.880 and yields an eigen value of 5.471 for the first component, indicating that 39.075% of the variance is accounted by this component (Table 6). For the second component, the internal consistency coefficient is 0.602 with an eigen value of 2.269, indicating that its proportion of variance is 16.204%. On the whole, the internal consistency coefficient (Cronbach’s Alpha) for the bi-dimensional model was 0.938, and the eigen value yielded of 7.739, explaining a total of 55.279% of the variability.

Table 6 shows the internal consistency coefficients (Cronbach’s Alpha), eigenvalues and percentage of variability explained by each of the components of the three and four-dimensional models. On the whole, the internal consistency coefficient (Cronbach’s Alpha) for the three and four-dimensional models were 0.961 and 0.976, respectively. The eigen value yielded for the three and four-dimensional models were of 9.301 and 10.627, respectively, and they explained a total of 66.435% and 75.910% of the variability, respectively.

### 3.2. Model and Operant Conditioning Test Behavioral Variability Explanatory Quality

CATREG R squared coefficient obtained ranged from 0.288 to 0.375 for the response type, and response intensity and learning ability variables, respectively (Table 4). In the same way, when CATPCA was implemented, four and three-dimensional models accounted for 75.910% and 66.435% of the total variance of behavioral variables, respectively. These results could compare to those obtained by CATREG. These findings address the fact that two of the components of the study could be summarized into one, with a low loss (9.475%) in the explanatory power of the variability. This low loss could stem from the fact that the response type variable was obtained classifying the levels in the mood/emotion variable, so that response type variable somehow derived from the mood/emotion variable. This percentage of loss is around the same value shown by CATPCA for the explanatory power of the 4th dimension (11.280%).

## 4. Discussion

Our statistical outputs suggest that the operant conditioning tests and model designed and used for our study efficiently and successfully enable quantifying the variation in the adaptive and cognitive behavioral response of donkeys (Table 4 and Table 7).

Cramer’s V has been stated to be the most suitable parameter for assessing factor strength and testing for significance after the results of cross-sectional studies relying on chi-square analyses. Although most meteorological or climatological variables could be assumed to be approximately normally distributed, some other such as rainfall, remarkably deviate from a Gaussian distribution [18]. Chi-square tests become then especially relevant, as they are neutral to the parametric or non-parametric nature of the distribution and relatively robust to situations in which there are only a limited number of data common to endangered populations, as it would be the case of donkey breeds.

As our results suggest, when we aim at comparing continuous environmental factors relying on linear scales with accurately described behavioral or cognitive categorical variables, it is useful to homogenize their nature, turning continuous variables into categorical ones. This homogenization may simplify establishing effective, easily-understandable relationships.

According to Cohen [19], when using Cramer’s V, small effect associations may range from 0.0 to 0.10, medium effect associations from 0.3 to 0.5 and large effect associations from 0.5 to anything above. The same author would suggest this parameter to be especially suitable for behavioral or psychological studies, considering a statistically significant measure of *p* < 0.05 with a smaller or greater effect size to indicate a meaningful difference among the categories of a particular factor influencing the different categorical levels of the variables under study.

While studying our first hypothesis, Chi-Square and Cramér’s V highlighted there was a significant linear correlation between environmental factors and variables (Table 3), although the behavioral variables tested were not dependent on some of them as shown in the result section.

Chi-Square and Cramér’s V highlighted there was a highly statistically significant linear correlation (*p* < 0.001) between all environmental factors and variables, except for season at birth which was just significant (*p* < 0.05) for response intensity and learning ability and non-significant for response type (Table 3). However, the only factor behavioral variables tested were not dependent on some of them as shown in the result section.

Date of birth has been extensively reported to influence behavior and cognitive abilities in animal models which have later been applied to humans [20,21] with an underneath basis relying on circadian rhythms [22], frequently or exclusively focusing on the influence of birth months. However, the CATREG standardized coefficients and CATPCA component loadings reported found an almost three times lower variation and therefore a weaker factor strength for the birth season when compared to birth year. This low variation among seasons could rely on season shifting, one of the most widely discussed events of climate change [23]. The occurrence of shifting seasons is directly linked to warmer worldwide temperatures. According to Stine et al. [24], the amplitude component of the annual cycle (half the difference between summer and winter temperatures) has progressively decreased in most continental areas. This situation translates into the occurrence of warmer winters resulting in a lower seasonal weather variation through the year, as our results suggest. In the same way, the greater importance and higher relative frequency for birth year variations may support all of the long-term progressively increasing temperature records existing from one year to another since 1884 [25].

It may be worth noting that the late gestation of the animals displaying a depressive behavior pattern took place during the winter to early spring of 2005, when the cold wave accounting for the lowest temperature in the last 117 years, took place in Spain [26]. This situation may be worsened given the characteristics of the light grey coat of Andalusian donkeys which makes them more sensitive to cold weather. Furthermore, the animals born during that spring were all jennies. Studies in humans [27] and rats [28] have reported that the pregnancies of mothers who had been exposed to extreme weather conditions not only presented a resulting offspring with a lower weight at birth and at increased risk to experience developmental, learning, and emotional disorders, but also an altered sex ratio, lowering the occurrence of newborn male offspring in different species [29,30,31].

Moon phases have been reported to increase the number of deliveries in cows [32]. The same authors would report that apart from the higher birth rates of the dairy cows near and during the full moon, the predicted and real delivery dates significantly differed within the eight moon phases. Cows with predicted delivery dates before the first-quarter moon tended to deliver later than expected, whereas cows with delivery dates on a full moon to last-quarter phase tended to deliver on schedule. Although our study is the first to attempt the assessment of the effect of the moon phase at birth on mood or behavior, it is possible that this reported alteration on the times at delivery may be the basis for different degree alterations of cognitive development. These cognitive alterations may translate into future behavioral mood statuses, as suggested by the near 10% linearly correlated effect of moon phase at birth on learning abilities and 12% linearly correlated effect of moon phase at birth on mood found in our study through Cramér’s V and CATREG. Figure 1 shows the relative frequency distribution for different mood/emotion patterns displayed by the donkeys relative to the phase of the moon at the moment of birth and at the time of evaluation.

Our results support the information found by Zakari et al. [33], according to which the behavioral repertoire of donkeys is modulated depending on the season. This seasonal evaluation effect has also been reported by equid welfare organizations such as The Brooke in working donkeys [34]. The study by Meyer et al. [35] in humans reported cognitive abilities to be distorted by a seasonal effect linked to serotonin levels in humans with better cognitive performance in summer, what extended to our experience could explain the increased frequency of animals refusing to cross the unknown surface. Donkeys’ increased cognitive abilities have been mistaken with stubbornness. Therefore, refusal to cross new surfaces may be related with an increased ability to assess potentially harmful or dangerous situations.

Moon phase has been reported to alter both humans and animal at many different psychological and physiological levels [6]. A slight decrease in the strength of the effect of moon phase at the date of evaluation of more than half the strength for the effect of moon phase at birth was reported according to CATREG standardized coefficients. Cramér’s V for moon phase at evaluation was around half the value for moon phase at birth, what suggested a stronger linear correlation between this factor and mood, response intensity, and learning ability variables. The power that the moon exerts on living beings may be mainly attributed to two factors or primary forces which differ along the consecution of the moon cycle; gravity and light changes, and their suggested effect on hormonal production and regulation. Folklore has reported a possibly calmer, hyporeactive status and low cognitive abilities in marine animals like the whale shark, which, as South Sea Islanders believe, are most easily caught a few days after a full moon. In the same way, the Miskito Indians of Eastern Nicaragua, believe that all animals respond to tides, that the woodpecker pecks when the tide is changing, and that hunting and fishing are best at the rising tide, but not at a new moon [36]. This has also been reported for hunting behavior in such large felines as lions, which were prone to hunt larger preys during new moon phases [37]. The time between two successive high or low tides is 12.4. A “lunar day” is 24.8 h. Tides are greatest at a new moon when the gravitational pull of the sun and moon are both acting in the same direction. Because the moon is moving relative to the Earth and the Sun, “lunar days” are not precisely 24 h [38], which at the same time alters normal light cycles. LeGates et al. [39] reported that when subjected to an abnormal light cycle, mice’s cognitive and mood functions were directly affected through intrinsically photosensitive retinal ganglion cells, which may support the strength of the effects obtained for all variables in our study. The effect of the number of sunlight hours found in our study not only was the stronger one according to CATREG standardized coefficients but also the one holding the strongest inverse correlations for all the dimensions in the CATPCA. Exposure to unnatural lighting can induce significant changes in affection, increasing depressive-like and decreasing anxiety-like responses as it disrupts circadian rhythms of locomotor activity, body temperature, hormones, and the sleep-wake cycle in animals [40].

Behavioral responses and mood have been reported to be altered because of weather conditions and the effects of high and low extreme temperature and relative humidity, although still no previous study assessing the direct correlation with weather conditions or environment temperature has been carried out. The results by Denissen et al. [41] revealed the main effects of temperature, wind power, and sunlight on negative emotion patterns in humans and this could be extrapolated to donkeys as highlighted by the CATPCA loadings and CATREG standardized coefficients observed for the temperature, relative humidity and weather conditions on the four variables tested (Table 3, Table 5, Table 8 and Appendix A). The basis for this behavioral and possibly cognitive repercussion could be, as stated by [42], the fact that endothermic animals such as equids usually keep their body temperature within narrow limits with changing environmental conditions in an attempt to cool brain temperature. This advantage means a drawback as well, as it occurs at a high energetic cost, making endothermic animals face a two-fold challenge. This double challenge could be one of the reasons, as reported by Janczarek et al. [43], for adverse changes in the behavior of recreational horses that can occur if the horse is ridden when the air temperature is above 26 °C. These conditions may cause an alteration in mood, with donkeys showing more elusive and hyporeactive responses, and a reduction in the willingness to work in horses and other equids. In our study, this was supported by the increase on the refusal to cross and lack of cooperation when completing the problem-solving test, a decrease on the frequency for neutral responses and an increase in the frequencies for rejective and fearful attitudes when temperature ranged from 25 to 29 °C.

Relative humidity has been reported to be a thermally stressing factor from a welfare perspective and to affect donkey behavior and performance when it reaches extreme upper values as reported by Zakari et al. [5] and Gebresenbet et al. [44]. Heat loss mechanisms include evaporation, skin blood flow, and cardiovascular support for thermoregulation and exercise. Low temperatures have been reported to inhibit sweat gland in the donkey [45] and when simultaneously relative humidity is high this effects increase. Sweat does not readily evaporate from the body, and therefore it cannot reduce its temperature efficiently. When this rate is low, such evaporation rate is excessive therefore causing mucosa and skin dryness and increasing heart rate [44]. This situation alters performance in working donkeys and has been reported to reduce complex cognitive capacities in humans [46]. Parallelly, the low cooperative response frequency may be attributed to the fact that as temperature increases and relative humidity decreases, when kept around an optimal point for donkeys, they may be prone to display natural behaviors. Donkeys are energetic natural savers [47], and they will tend to slow moving and decrease their behavioral activity rather than display the compensative methods that they are likely to present under stressing meteorological situations [5].

Extreme high windspeed has been reported to be a welfare distorting factor for donkeys [5,48] to which individuals may adapt differently. Interestingly, as windspeed decreased, the responses of the donkeys became milder, and their attitudes turn less cooperative. White or light coat animals such as the Andalusian donkey have been reported to absorb more heat under higher to 3 m/s windspeeds, which may make them develop more stressful responses [49], hence, the high frequency for stress related moods and slightly lower intensity responses for calmer or cooperative moods. The low variation found, may account for the similar values obtained for almost all the variables. Similarly to our findings, studies in mice have reported a pronounced behavioral inhibition as well as a cognitive disruption because of an increase in the duration of light phases per day, which should be considered when testing animals for such traits [50].

Slight barometric pressure fluctuations have traditionally been reported to promote behavioral and feeding activity in fish. Fishers usually relate slight changes towards high pressure to clear sky occurrence during which fishing is medium to slow as fish may slowly be in deeper water or near cover. These trends progressively invert when there is falling pressure, the best attributed timing for fishing during degrading weather when fish are more active what may support our results [51], though still no previous scientific research has been carried out on the effects of slight variations on barometric pressure. Studies on rats have reported individuals to be more prone to develop depressive behavioral patterns when they are exposed to a sharp fall in barometric pressure (20 hPa below the natural atmospheric pressure) [52]. However, the animals in our study were not exposed to such extreme air pressure variations.

Rainfall has been reported to be especially crucial as a welfare distorting or stressful provoking factor in donkeys [48]. Curiously, donkeys have traditionally been attributed the ability to predict lousy weather (Graphical abstract) and rain occurrence [53,54] as it could be stated by this study, although this may be the first attempt to scientifically proof such ability.

## 5. Conclusions

Environmental conditions, seasonal, timing (year) and moon cycle phases are potential stress factors or behavioral modulators that affect the behavior and cognitive responses of donkeys, as well as may have potentially long lasting effects which can be traced back. Climate oscillation effects may affect donkeys altering their physiological biorhythms and produce severe behavioral and cognitive modifications. Deviations in behavioral patterns or on the abilities of the donkeys to perform complex tasks to which they may not be accustomed may become relevant indicators of welfare as well as they may address the most suitable techniques or methods to be applied in each case. Furthermore, behavior becomes a relevant tool when predicting future weather conditions as well as may report the potential distortion that they may cause, a prominent importance fact for veterinarians, practitioners and donkey owners, as it may allow them to anticipate such situations in order to counteract their effects.

## Figures and Tables

**Figure 1 animals-08-00215-f001:**
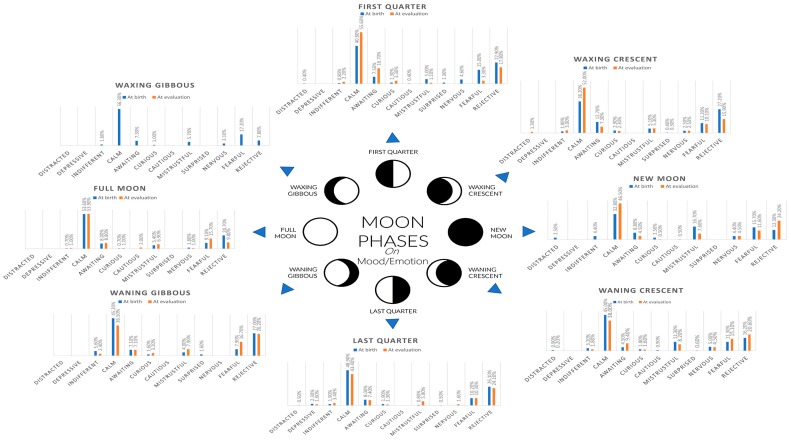
Relative frequency distribution for different mood/emotion patterns displayed by the donkeys relative to the phase of the moon at the moment of birth and at the time of evaluation.

**Table 1 animals-08-00215-t001:** Description of the operant conditioning test used in the study.

Test Factors	Descriptions
Time per stage/treatment presentation	75 s per stage/treatment presentation. The application of the reinforcement treatments that Handler A, Handler B or both implemented to lead the donkey across the oilcloth lasted for the whole 75 s. These treatments were applied to check the response of the animals to the different types of reinforcement. No additional time was supplied for the donkeys to complete the stages, so that, once the 75 s, provided to the donkeys to interact with the elements presented, had expired, the following stage started and the next treatment was implemented.
Test duration	450 s.
Test stages	1 to 6. Each stage corresponded to the implementation of each of the six reinforcement treatments.
Previous considerations	The oilcloth was the element (obstacle) that the donkeys were led to cross over. No donkey had been in contact with the oilcloth previous to the test. Handlers A and B, used 6 reinforcement treatments to lead the donkeys cross over such obstacle.The donkeys were accustomed to the area in which the test took place as it was an open area on which the donkeys used to carry out their daily activities.The donkeys that were taking the test were not present while the oilcloth was being laid on the floor for the first time. The donkeys were assessed one at a time, so no additional donkey was present while the test was taking place.The test started when Handler B raised the oilcloth and relayed it again on the floor in front of the donkey being tested. This action only took place 1 minute before stage 1 (before the 1st treatment was implemented) and was not repeated further in the test. Cameraman started controlling time after the oilcloth had been relayed, when Handler A gave the first step forward towards the oilcloth.Frontal and visual elements fell within the visual scope of the donkeys, while we considered rear elements those that fell into a blind area. Acoustic elements could be frontal or rear and emitted sounds.Reinforcement treatments comprised different elements. Known elements were those which had already been presented to the donkeys at any point in their lives (relying on owner’s information), while unknown elements were those to which, according to the owner, the donkeys were not acquainted.All the reinforcement treatments were implemented sequentially and consecutively from stage 1 to 6, one after another, without any stop between each of them, whether the donkey had completed each stage (crossed the obstacle) completely or not (avoided it). That is to say, the fact that an animal crossed/avoided the oilcloth completely in one of the treatments from 1 to 6, did not prevent the rest of treatments from being implemented.
Legend	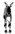	Donkey being tested.
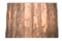	2 × 2 m oilcloth with a wooden print.
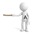	Rope leader/Handler A.
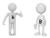	Handler B/Lurer (in Stage 4)/2nd Rope leader (in Stage 5)/Clapper (in Stage 6).
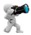	Cameraman (C)/Time controller.
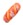	Treat (bread, carrots, feed or sugar lumps). Carried by Handler B.
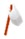	Motivator. Plastic bag attached to a wooden stick. Carried by Handler B.
**Test Stage**	**Descriptions**	**Test Stage**	**Descriptions**
STAGE 1 (S1)Treatment 1: Soft voice 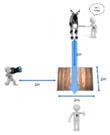	Oilcloth presented to the donkey for the first time (Frontal unknown element).The donkey is given 75 s to complete Stage 1, that is to cross over the oilcloth.Using a lead rope and soft voice, Handler A tried to comfort the donkey to make it cross the oilcloth on the floor, but without pulling from the rope if the donkey refused to move (Neutral reinforcement).	STAGE 2 (S2)Treatment 2: Pressure to leading rope 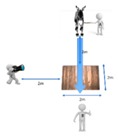	Donkey had already had contact with the oilcloth in Stage 1 (Frontal known element).Using a lead rope with applied pressure to make the donkey cross over the oilcloth. Handler A released the pressure when the donkey moved to cross the oilcloth (Negative reinforcement).
STAGE 3 (S3)Treatment 3: Treat 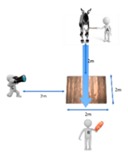	Donkey had already had contact with the oilcloth in Stage 1 and 2 and was familiar to the treat given (Frontal known elements).Handler B offered a familiar treat to lead the donkey to cross over the oilcloth (the treat offered depended on the owner’s tastes and therefore the animals were familiar to it. Handler B used the treat that the owner of each donkey normally offered them to tease them. All animals did not accept any other treat that had not been offered to them by their owners previous to the test, as the field experiences reported) (Positive reinforcement/Luring).	STAGE 4 (S4)Treatment 4: Motivator 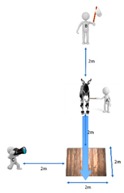	Donkey had already had contact with the oilcloth in Stage 1, 2 and 3 (Frontal known element and rear unknown element).Handler A applied pressure to the lead rope at the same time Handler B made a noise from behind the donkey with a so-called “donkey motivator” (plastic bag tied on the end of a stick. The donkey was led by slightly pulling the rope until it crossed the oilcloth completely (Negative reinforcement).
STAGE 5 (S5)Treatment 5: Double rope leading 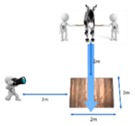	Donkey had already had contact with the oilcloth in Stage 1, 2, 3 and 4 (Frontal known element).Using two lead ropes attached on either side of the halter, Handlers A and B encouraged the donkey across, releasing the pressure when the donkey moved and then reapplied when it stopped until it crossed the oilcloth completely (Negative reinforcement).	STAGE 6 (S6)Treatment 6: Clapping 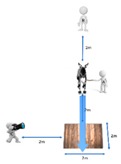	Donkey had already had contact with the oilcloth in Stage 1, 2, 3, 4 and 5 (Frontal and rear known elements).Handler B clapped his hands from behind the donkey to make it move forward. Handler A applied pressure on the lead rope and while the donkey was led across by the auditory sound of the claps, pressure and sound were released or stopped when the donkey moved and reapplied when it stopped until the donkey had completed the task (Negative reinforcement).

**Table 2 animals-08-00215-t002:** Description of the treatments and stimuli presented, their reinforcement classification and terminology considered.

Treatment/Stimulus	Stimulus Description	Stimulus Type	Reinforcement
Treatment 1 (S1): Soft voice	Handler (B) uses a lead rope and soft voice, trying to comfort the donkey to make the donkey cross the oilcloth on the floor, but without pulling the rope if the donkey refuses to move.	Unknown frontal visual stimulus.	Neutral ^a^
Treatment 2 (S2): Pressure to leading rope	Handler (B) uses a lead rope with applied pressure to make the donkey cross over the oilcloth. Handler (B) releases the pressure when the donkey moves as it crosses the oilcloth.	Known frontal visual stimulus.	Negative ^b^
Treatment 3 (S3): Treat	A familiar treat is used to lure the donkey (dry bread, carrots or feed, depending on the owner’s tastes and to which the donkeys on each farm were accustomed). We use the treat that the owner regularly uses as a treat for all of the donkeys in the same farm (the attraction or attention of the animals to the treats depends on whether they are used to the treats presented or not as empirical observations had revealed at a preliminary stage when developing the operant conditioning test). When the donkeys are not familiar to the treats presented, they do not respond to the stimulus by handler (C). The treat is given to the donkey once the task is completed.	Known frontal visual stimulus.	Positive/Luring ^c^
Treatment 4 (S4): Motivator	Handler (B) applies pressure to the lead rope, and handler C makes noise from behind the donkey with a so-called “donkey motivator” (plastic bag tied on the end of a stick) [12]. Handler (B) leads the donkey by slightly pulling the rope until the donkey crosses the oilcloth completely.	A known frontal visual stimulus and an unknown rear auditory stimulus.	Negative
Treatment 5 (S5): Double rope leading	Two handlers (B and C) using two lead ropes attached on either side of the halter to encourage the donkey across. The handlers (B and C) release the pressure when the donkey moves and then reapply the pressure when it stops until the donkey crosses the oilcloth completely.	Known frontal visual stimulus.	Negative
Treatment 6 (S6): Clapping	Handler (B) applies pressure on the lead rope, and handler (C) encourages the donkeys across by an acoustic sound. Handler C claps their hands from behind the donkey to make it move forward [13]. Pressure and sound are released or stopped when the donkey moves and reapplied when it stops until the donkey had completed the task.	A known frontal visual stimulus and an unknown rear acoustic stimulus.	Negative

A full description of the protocols, scales, and methods used in this study is described in Navas et al. [9] and Navas González et al. [11]. The terminology used to classify stimuli throughout this paper rests on classical concepts, as applied by Sankey et al. [14]. According to these authors stimuli can be perceived as negative, neutral or positive. ^a^ Neutral reinforcement training implies the donkey perceives the tasks to be neither positive nor aversive and therefore the stimulus does not act to reinforce or punish the donkey’s behavior. Therefore, the animal fails to respond to the stimuli and continues quietly and calmly with the task uninterrupted [15]. ^b^
*Negative reinforcement* implies delivering an unpleasant stimulus and terminating it when an individual performs a presented task in the desired manner or expresses the desired behavior [16]. ^c^
*Positive/luring reinforcement* implies the presentation of a pleasant stimulus (lure) when an individual fulfils a task in the desired manner or expresses the desire and the behavior [16].

**Table 3 animals-08-00215-t003:** Statistical significance and strength of the effects on the different variables tested in donkeys in this study.

Variable	N	Response Type			Mood/Emotion			Response Intensity			Learning Ability		
		χ^2^	*p*-Value	Cramer’s V	χ^2^	*p*-Value	Cramer’s V	χ^2^	*p*-Value	Cramer’s V	χ^2^	*p*-Value	Cramer’s V
**Environmental/Meteorological**													
Year of evaluation	1800	76.99	<0.001 ***	0.146	256.34	<0.001 ***	0.267	138.40	<0.001 ***	0.196	138.40	<0.001 ***	0.196
Season of evaluation	1800	70.54	<0.001 ***	0.198	114.27	<0.001 ***	0.252	49.60	<0.001 ***	0.166	49.60	<0.001 ***	0.166
Weather conditions	1800	16.71	<0.001 ***	0.096	87.12	<0.001 ***	0.220	77.51	<0.001 ***	0.208	77.51	<0.001 ***	0.208
Temperature	1800	81.46	<0.001 ***	0.150	152.10	<0.001 ***	0.206	136.99	<0.001 ***	0.195	136.99	<0.001 ***	0.195
Moon phase at evaluation	1800	50.52	<0.001 ***	0.118	159.28	<0.001 ***	0.121	66.72	<0.001 ***	0.096	66.72	<0.001 ***	0.096
Relative humidity	1800	49.39	<0.001 ***	0.117	275.41	<0.001 ***	0.226	56.35	<0.001 ***	0.102	56.35	<0.001 ***	0.102
Windspeed	1800	146.78	<0.001 ***	0.202	332.77	<0.001 ***	0.248	178.81	<0.001 ***	0.182	178.81	<0.001 ***	0.182
Sunlight hours	1800	135.56	<0.001 ***	0.194	271.25	<0.001 ***	0.274	266.23	<0.001 ***	0.272	266.23	<0.001 ***	0.272
Barometric pressure	1800	109.42	<0.001 ***	0.174	362.36	<0.001 ***	0.317	189.71	<0.001 ***	0.230	189.71	<0.001 ***	0.230
Rainfall per day	1800	112.73	<0.001 ***	0.177	325.54	<0.001 ***	0.301	221.94	<0.001 ***	0.248	221.94	<0.001 ***	0.248
Rainfall on the following day	1800	121.10	<0.001 ***	0.183	373.48	<0.001 ***	0.263	224.46	<0.001 ***	0.204	224.45	<0.001 ***	0.204
**Animal Birth**													
Season of birth	1800	6.88	0.194	0.049	80.90	<0.001 ***	0.122	34.12	<0.05 *	0.079	34.12	<0.05 *	0.079
Year of birth	1800	347.07	<0.001 ***	0.310	875.91	<0.001 ***	0.210	265.58	<0.001 ***	0.192	265.58	<0.001 ***	0.192
Moon phase at birth	1800	44.75	<0.001 ***	0.111	270.38	<0.001 ***	0.388	77.86	<0.001 ***	0.208	77.85	<0.001 ***	0.208

Levels of significance are indicated by * and *** for *p* < 0.05, statistically significant and *p* < 0.001, highly statistically significant, respectively.

**Table 4 animals-08-00215-t004:** Model summary of stepwise linear regression with transformed variables.

Variable	R	R Square	Adjusted R Square	Significance
Response type	0.537	0.288	0.265	<0.001
Mood/emotion	0.608	0.370	0.350	<0.001
Intensity of response	0.612	0.375	0.355	<0.001
Learning ability	0.612	0.375	0.355	<0.001

**Table 5 animals-08-00215-t005:** Standardized coefficients and significance of categorical regression (CATREG) model.

Variable	Response Type		Mood/Emotion		Response Intensity		Learning Ability	
	β	Sig.	β	Sig.	β	Sig.	β	Sig.
Year of birth	0.235	<0.001	0.212	<0.001	0.195	<0.001	0.195	<0.001
Season of birth	0.053	<0.001	0.075	<0.001	0.054	<0.001	0.054	<0.001
Relative humidity	0.136	<0.001	0.263	<0.001	0.106	<0.001	0.106	<0.001
Year of evaluation	0.196	<0.001	0.242	<0.001	0.065	0.031	0.065	0.042
Season of evaluation	0.129	0.058	0.116	0.113	0.621	<0.001	0.621	<0.001
Weather conditions	0.121	0.001	0.211	<0.001	0.029	0.257	0.029	0.267
Temperature	0.206	<0.001	0.230	<0.001	0.040	0.230	0.040	0.244
Moon phase at birth	0.098	<0.001	0.117	<0.001	0.093	<0.001	0.093	<0.001
Moon phase at evaluation	0.145	<0.001	0.111	<0.001	0.107	<0.001	0.107	<0.001
Windspeed	0.304	<0.001	0.395	<0.001	0.280	<0.001	0.280	<0.001
Sunlight hours	0.527	<0.001	0.596	<0.001	0.814	<0.001	0.814	<0.001
Barometric pressure	0.285	<0.001	0.365	<0.001	0.054	0.115	0.054	0.130
Rainfall on that day	0.166	0.044	0.103	0.105	0.231	0.013	0.231	0.011
Rainfall on the following day	0.387	<0.001	0.468	<0.001	0.670	<0.001	0.670	<0.001

β = Standardized coefficients; Sig. = Significance.

**Table 6 animals-08-00215-t006:** CATPCA model summary.

Dimension	Cronbach’s Alpha	Total (Eigenvalue)	% of Variance	Dimension	Cronbach’s Alpha	Total (Eigenvalue)	% of Variance	Dimension	Cronbach’s Alpha	Total (Eigenvalue)	% of Variance
1	0.849	4.733	33.804	1	0.876	5.351	38.225	1	0.880	5.471	39.075
2	0.618	2.347	16.767	2	0.594	2.228	15.914	2	0.602	2.269	16.204
3	0.530	1.968	14.058	3	0.451	1.721	12.296				
4	0.395	1.579	11.280								
Total	0.976 ^a^	10.627	75.910	Total	0.961 ^a^	9.301	66.435	Total	0.938 ^a^	7.739	55.279

^a^ Total Cronbach’s Alpha is based on the total eigenvalue.

**Table 7 animals-08-00215-t007:** Categorical principal component analyses (CATPCA) component loadings.

Environmental Factors	Dimension		Environmental Factors	Dimension			Environmental Factors	Dimension			
	1	2		1	2	3		1	2	3	4
Rainfall on the following day	**0.974**	0.127	Season	**0.974**	0.167	0.032	Rainfall on the following day	**0.964**	0.046	−0.209	0.115
Sunlight hours	**−0.974**	−0.148	Sunlight hours	**−0.973**	−0.180	−0.037	Rainfall per day	**0.964**	0.044	−0.211	0.116
Season	**0.973**	0.123	Rainfall on the following day	**0.972**	0.184	0.036	Sunlight hours	**−0.954**	−0.110	0.207	−0.149
Rainfall per day	**0.972**	0.132	Rainfall per day	**0.971**	0.188	0.039	Season	**0.954**	0.080	−0.225	0.139
Year of evaluation	**0.754**	0.142	Year of evaluation	**0.745**	0.052	0.370	Barometric pressure	**0.703**	0.155	**0.574**	−0.161
Barometric pressure	**0.666**	−0.372	Barometric pressure	**0.651**	−0.349	−0.220	Year of evaluation	0.183	**0.700**	**0.601**	0.017
Temperature	−0.448	−0.206	Temperature	−0.437	−0.350	0.377	Windspeed	−0.405	**0.694**	−0.33	0.476
Windspeed	−0.344	**0.871**	Windspeed	−0.336	**0.810**	0.342	Relative humidity	−0.489	**0.660**	−0.312	0.453
Relative humidity	−0.404	**0.846**	Relative humidity	−0.474	**0.738**	0.353	Temperature	0.274	**0.610**	0.329	0.035
Season of birth	0.095	−0.424	Season of birth	0.125	−0.444	0.180	Season of birth	0.149	−0.353	0.264	0.246
Year of birth	−0.331	−0.363	Moon phase at birth	0.068	0.375	−0.046	Weather conditions	−0.141	0.291	**0.634**	−0.198
Moon phase at birth	0.070	0.360	Year of birth	0.075	−0.436	**0.659**	Moon phase at evaluation	−0.075	−0.324	0.323	**0.659**
Moon phase at evaluation	−0.180	−0.220	Moon phase at evaluation	−0.008	−0.336	**0.589**	Year of birth	0.001	−0.329	0.392	**0.622**
Weather conditions	−0.173	−0.179	Weather conditions	−0.189	−0.095	**−0.576**	Moon phase at birth	0.002	−0.314	0.23	0.362

Numbers in bold highlight meaningfully contributing factors to each model (>|0.5|).

**Table 8 animals-08-00215-t008:** Regression equations for the behavioral variables assessed.

Model	Regression Equation	Legend
General model	Z’y_tmil_ = β_RainfallPrediction_Z_RainfallPrediction_ + β_Sunlighthours_Z_Sunlighthours_ + β_Season_Z_Season_ + β_Rainfall_Z_Rainfall_ + β_Year_Z_Year_ + β_BarometricPressure_Z_BarometricPressure_ + β_Temperature_Z_Temperature_ + β_Windspeed_Z_Windspeed_ + β_Relativehumidity_Z_Relativehumidity_ + β_BirthSeason_Z_BirthSeason_ + β_BirthYear_Z_BirthYear_ + β_BirthMoon_Z_BirthMoon_ + β_Moonphase_Z_Moonphase_ + β_Weather_Z_Weather_	Z’y_tmil_ = Z score for each behavioral categorical variable (Response type, response intensity, mood/emotion and learning ability). β = standardized coefficient for each of the noncognitive categorical factors appearing in the subindex. Z = Z score for each of the noncognitive categorical factors appearing in the subindex.
Response type	Z’y_t_ = 0.387(Z_RainfallPrediction_) + 0.527(Z_Sunlighthours_) + 0.166(Z_Rainfall_) + 0.196(Z_Year_) + 0.285(Z_BarometricPressure_) + 0.206(Z_Temperature_) + 0.304(Z_Windspeed_) + 0.136(Z_Relativehumidity_) + 0.053(Z_BirthSeason_) + 0.235(Z_BirthYear_) + 0.098(Z_BirthMoon_) + 0.145(Z_Moonphase_) + 0.121(Z_Weather_)	Z’y_t_ = Z score for response type variable. β_RainfallPrediction_Z_RainfallPrediction_ = 0.387(Z_RainfallPrediction_) β_Sunlighthours_Z_Sunlighthours_ = 0.527(Z_Sunlighthours_) β_Rainfall_Z_Rainfall_ = 0.166(Z_Rainfall_) β_Year_Z_Year_ = 0.196(Z_Year_) β_BarometricPressure_Z_BarometricPressure_ = 0.285(Z_BarometricPressure_) β_Temperature_Z_Temperature_ = 0.206(Z_Temperature_) β_Windspeed_Z_Windspeed_ = 0.304(Z_Windspeed_) β_Relativehumidity_Z_Relativehumidity_ = 0.136(Z_Relativehumidity_) β_BirthSeason_Z_BirthSeason_ = 0.053(Z_BirthSeason_) β_BirthYear_Z_BirthYear_ = 0.235(Z_BirthYear_) β_BirthMoon_Z_BirthMoon_ = 0.098(Z_BirthMoon_) β_Moonphase_Z_Moonphase_ = 0.145(Z_Moonphase_) β_Weather_Z_Weather_ = 0.121(Z_Weather_)
Mood/Emotion	Z’y_m_ = 0.468(Z_RainfallPrediction_) + 0.596(Z_Sunlighthours_) + 0.242(Z_Year_) + 0.365(Z_BarometricPressure_) + 0.230(Z_Temperature_) + 0.395(Z_Windspeed_) + 0.263(Z_Relativehumidity_) + 0.075(Z_BirthSeason_) + 0.212(Z_BirthYear_) + 0.117(Z_BirthMoon_) + 0.111(Z_Moonphase_) + 0.211(Z_Weather_)	Z’y_m_ = Z score for the mood/emotion variable. β_RainfallPrediction_Z_RainfallPrediction_ = 0.468(Z_RainfallPrediction_) β_Sunlighthours_Z_Sunlighthours_ = 0.596(Z_Sunlighthours_) β_Year_Z_Year_ = 0.242(Z_Year_) β_BarometricPressure_Z_BarometricPressure_ = 0.365(Z_BarometricPressure_) β_Temperature_Z_Temperature_ = 0.230(Z_Temperature_) β_Windspeed_Z_Windspeed_ = 0.395(Z_Windspeed_) β_Relativehumidity_Z_Relativehumidity_ = 0.263(Z_Relativehumidity_) β_BirthSeason_Z_BirthSeason_ = 0.075(Z_BirthSeason_) β_BirthYear_Z_BirthYear_ = 0.212(Z_BirthYear_) β_BirthMoon_Z_BirthMoon_ = 0.117(Z_BirthMoon_) β_Moonphase_Z_Moonphase_ = 0.111(Z_Moonphase_) β_Weather_Z_Weather_ = 0.211(Z_Weather_)
Response intensity	Z’y_i_ = 0.670(Z_RainfallPrediction_) + 0.814(Z_Sunlighthours_) + 0.621(Z_Season_) + 0.231(Z_Rainfall_) + 0.065(Z_Year_) + 0.280(Z_Windspeed_) + 0.106(Z_Relativehumidity_) + 0.054(Z_BirthSeason_) + 0.195(Z_BirthYear_) + 0.093(Z_BirthMoon_) + 0.107(Z_Moonphase_)	Z’y_i_ = Z score for the response intensity variable. β_RainfallPrediction_Z_RainfallPrediction_ = 0.670(Z_RainfallPrediction_) β_Sunlighthours_Z_Sunlighthours_ = 0.814(Z_Sunlighthours_) β_Season_Z_Season_ = 0.621(Z_Season_) β_Rainfall_Z_Rainfall_ = 0.231(Z_Rainfall_) β_Year_Z_Year_ = 0.065(Z_Year_) β_Windspeed_Z_Windspeed_ = 0.280(Z_Windspeed_) β_Relativehumidity_Z_Relativehumidity_ = 0.106(Z_Relativehumidity_) β_BirthSeason_Z_BirthSeason_ = 0.054(Z_BirthSeason_) β_BirthYear_Z_BirthYear_ = 0.195(Z_BirthYear_) β_BirthMoon_Z_BirthMoon_ = 0.093(Z_BirthMoon_) β_Moonphase_Z_Moonphase_ = 0.107(Z_Moonphase_)
Learning ability	Z’y_i_ = 0.670(Z_RainfallPrediction_) + 0.814(Z_Sunlighthours_) + 0.621(Z_Season_) + 0.231(Z_Rainfall_) + 0.065(Z_Year_) + 0.280(Z_Windspeed_) + 0.106(Z_Relativehumidity_) + 0.054(Z_BirthSeason_) + 0.195(Z_BirthYear_) + 0.093(Z_BirthMoon_) + 0.107(Z_Moonphase_)	Z’y_i_ = Z score for the learning ability variable. β_RainfallPrediction_Z_RainfallPrediction_ = 0.670(Z_RainfallPrediction_) β_Sunlighthours_Z_Sunlighthours_ = 0.814(Z_Sunlighthours_) β_Season_Z_Season_ = 0.621(Z_Season_) β_Rainfall_Z_Rainfall_ = 0.231(Z_Rainfall_) β_Year_Z_Year_ = 0.065(Z_Year_) β_Windspeed_Z_Windspeed_ = 0.280(Z_Windspeed_) β_Relativehumidity_Z_Relativehumidity_ = 0.106(Z_Relativehumidity_) β_BirthSeason_Z_BirthSeason_ = 0.054(Z_BirthSeason_) β_BirthYear_Z_BirthYear_ = 0.195(Z_BirthYear_) β_BirthMoon_Z_BirthMoon_ = 0.093(Z_BirthMoon_) β_Moonphase_Z_Moonphase_ = 0.107(Z_Moonphase_)

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
