# Peer review of "Can Donkey Behavior and Cognition Be Used to Trace Back, Explain, or Forecast Moon Cycle and Weather Events?"

_animals, 2018, doi:10.3390/ani8110215_

Round 1

Reviewer 1 Report

Dear Authors,

Thank you for your excellent description of donkey behavior in relationship to the environment. Actual scientific data showing examples of environmental conditions and how the conditions effect the donkey's behavior and reactions. Very novel and unique work that contribute to the much misunderstood donkey.

1. In the M&M, check the first sentence regarding jacks and wording.

2. Minor edit, in Table 7, the first column the commas need to be changed to period.

Thank you for the excellent illustrations demonstrating the tests to achieve the results. More work on donkey behavior and cognition is needed. 

Author Response

Thank you for your excellent description of donkey behavior in relationship to the environment. Actual scientific data showing examples of environmental conditions and how the conditions effect the donkey's behavior and reactions. Very novel and unique work that contribute to the much misunderstood donkey.

Thank you for the excellent illustrations demonstrating the tests to achieve the results. More work on donkey behavior and cognition is needed. 

Point 1: In the M&M, check the first sentence regarding jacks and wording.

Response 1: We rewrote the first sentence to follow reviewer suggestions. Our study sample comprised 78 Andalusian uncastrated jacks and 222 uncastrated jennies (n=300), born from 1990 to 2012 and officially registered in the national studbook of the Andalusian donkey breed.

Point 2: Minor edit, in Table 7, the first column the commas need to be changed to period.

Response 2: Commas in the first column of Table 7 were changed to periods.

Reviewer 2 Report

Row 99 – Please before describe WHAT you measures and ONLY after the statistical approach;

Row 113 – Which do you mean for “stimulus 7”. Describe a table of stimula before the statistics

Row 123 – Please define the reinforcement treatment

Table 1 – Please explain before why you selected these stages, what is the rational reasoning before to describe the stages

-          Important factor if the treat change (variable!!!) for each donkey experiment. Please specify each time what treat is used – specific owner. Treats are more convenient/gradite/good

-          Is part of a reinforcement treatement the treat too ?

Row 138 – please specify table S2 with respect to Table 2

Row 176 – It is better here to remind the definition fully

Row 177 – please define these response as a Marcalli scale

Row 192-210 – please do a table with all variables analysed by statistics with definition and numerical parametrization

Author Response

Response to Reviewer 2 Comments

Comments and Suggestions for Authors

Response to general comments:

English was checked by an ESOL Cambridge examination instructor in order to check for grammar and syntax mistakes. Changes can be observed through the text marked with traced changes.

Point 1: Row 99 – Please before describe WHAT you measures and ONLY after the statistical approach;

Response 1: The whole Material and Methods section was reorganized to follow the reviewer’s suggestion. The sections were rearranged to follow the order listed below.

2.1. Animal sample

2.2. Information registration

2.3. Categorical behavioural variables

2.4. Qualitative Behavioural Assessment

2.5. Noncognitive categorical factors

2.6. Meteorological and moon cycle records

2.7. Operant conditioning behavioural test

2.8. Test and scoring system reliability

2.9. Statistical analysis

2.10. Justification for Statistical tests

Point 2: Row 113 – Which do you mean for “stimulus 7”. Describe a table of stimula before the statistics

 Response 2: There was a number mistake, only six treatments/stimulus were presented and each of them corresponded with each of the stages in the operant conditioning test as it has been added to the text of the section entitled 2.8. Test and scoring system reliability. All the information regarding each of the stimulus /treatments is presented and described in Tables 1 and 2.

Point 3. Row 123 – Please define the reinforcement treatment

Response 3. Reinforce treatments, stimuli descriptions and classification and their constituting elements are provided in Tables 1 and 2.

Point 4. Table 1 – Please explain before why you selected these stages, what is the rational reasoning before to describe the stages

Response 4. The reason for the election of these stages was the fact that from stimulus 1 to 6 the methods were increasingly aversive as described in 2.7. section. We further clarified it by adding the following information: The operant conditioning behavioural test was carried out in an open area to which the donkeys were previously accustomed (it was part of the area over which the donkeys developed their daily activities). During the operant conditioning test, the donkeys were made cross over a 200×200 cm oilcloth with a wooden print on it using increasingly aversive reinforcement methods (from stimuli 1 to 6). We added Table 2 as well further clarifying the description of treatments and stimuli presented.

Point 5.  Important factor if the treat change (variable!!!) for each donkey experiment. Please specify each time what treat is used – specific owner. Treats are more convenient/gradite/good.

Response 5. The treats did not change from one donkey to another. As specified in Table 2, the donkey was lured by a familiar treat (dry bread, carrots or feed, depending on the owner’s tastes and to which the donkeys on each farm were accustomed. We used the treat that the owner regularly used as a treat for all of the donkeys in the same farm. The attraction or attention of the animals to the treats depended on whether they were used to the treats presented or not as revealed empirical observations at a preliminary stage when developing the operant conditioning test. When the donkeys were not familiar to the treats presented they did not respond to the stimulus) by handler (C). The treat was given to the donkey when the task was completed.

Point 6. Is part of a reinforcement treatement the treat too?

Response 6. Yes, treat it is a part of treatment 3, classified as positive reinforcement/luring as described in Table 1, 2 and the papers published by Navas et al. in 2017 and 2018.

Point 7. Row 138 – please specify table S2 with respect to Table 2

Response 7. We specified it and linked both tables with references.

Point 8. Row 176 – It is better here to remind the definition fully

Response 8. A definition for each of the behavioural variables has been provided and added to Supplementary Table 4.

Point 9. Row 177 – please define these response as a Marcalli scale

Response 9. A Mercalli scale definition for each of the behavioural variables has been provided and added to Supplementary Table 4.

Point 10. Row 192-210 – please do a table with all variables analysed by statistics with definition and numerical parametrization.

Response 10. Supplementary Table S3 was created as suggested by reviewer including descriptive statistics and numerical parametrization of all the variables analysed. Definitions were included according to a previous suggestion by the same reviewer to Supplementary Table S4.

All tables were renumbered and recited in the text following the order of appearance.

Reviewer 3 Report

The authors observed cognitive reactions of donkeys using operant conditioning test on 300 donkeys and quantified the effect power of environmental factors using CATPCA and CATREG. They found effect power ranged from 7.9% for birth season on learning and 38.8% for birth moon phase on mood. This reviewer recognized that this manuscript contains well-sophisticated statistics and surprising and innovative conclusions. However, the statistics are too technical for this reviewer to understand completely and the reviewer cannot evaluate whether these statistics are perfectly reasonable or not. Although the reviewer studied these protocols until the review-dead line, he could not follow all the statistics in this manuscript completely. Because the reviewer believes the validity of their statistics is the most important point in this manuscript, evaluation of the the statistics by the any other specialist(s) is needed.

Other points

- The reviewer think that it is better for laypeople to show the raw data and data distribution using histograms. Especially, the data of mood and the moon phase at birth is more specifically documented because it is one of the most important conclusions.

- They found the relationship between the moon phase and mood, but it is hard to discuss it because the tendency of relationship was not well documented. It is better to propose what phase of the moon and what type of the mood is related to each other.

Author Response

Response to Reviewer 3 Comments

Comments and Suggestions for Authors

The authors observed cognitive reactions of donkeys using operant conditioning test on 300 donkeys and quantified the effect power of environmental factors using CATPCA and CATREG. They found effect power ranged from 7.9% for birth season on learning and 38.8% for birth moon phase on mood. This reviewer recognized that this manuscript contains well-sophisticated statistics and surprising and innovative conclusions. However, the statistics are too technical for this reviewer to understand completely and the reviewer cannot evaluate whether these statistics are perfectly reasonable or not. Although the reviewer studied these protocols until the review-dead line, he could not follow all the statistics in this manuscript completely. Because the reviewer believes the validity of their statistics is the most important point in this manuscript, evaluation of the the statistics by the any other specialist(s) is needed.

Other points

Point 1.- The reviewer think that it is better for laypeople to show the raw data and data distribution using histograms. Especially, the data of mood and the moon phase at birth is more specifically documented because it is one of the most important conclusions.

Response 1. We prepared Figure 1 to represent the distribution of mood/emotions for both the moon phase at the moment of birth of the animals and at the moment of evaluation. It was not possible to prepare a histogram for every single effect as it may have resulted in an excessive increase in the material to present in the manuscript which is already extense.

Point 2.- They found the relationship between the moon phase and mood, but it is hard to discuss it because the tendency of relationship was not well documented. It is better to propose what phase of the moon and what type of the mood is related to each other.

Response 2. Figure 1 depicts the relationship between mood/emotion and moon phase at the moment of birth and at the moment of evaluation in separate histograms for each of the moon phases assessed, so that it is easier to understand what phase of the moon and what type of the mood is related to each other as suggested by reviewer.

Round 2

Reviewer 2 Report

The main problem of this paper is the following: it is not reproducible scientifically by a Galileian method, which is the essence of the SCIENCE. In particular the authors still not divide completely the numbers obtained by their manual introduction of the numbers in the experimental tables and the numbers obtained AFTER by the statistical treatments and factor analyses.

In summary this paper is not reproducible because is confused in the two different – very different – stage of the work. The first stage is made of “observations” reduced to “numbers” following some “normalization” criteria. The second stage is to introduce these EXPERIMENTALLY OBTAINED numbers in the statistical codes.

Therefore the paper should completely and definitively divide the Tables concerning the experimental phases and the tables concerning the statistical approach.

In other words, the reader/reviewer should imagine the complete and full reproduction of the experiment in other 22 farms i.e., in Italy or in Moldavia, in the same manner. I repeat SAME MANNER. From this paper is impossible because is lacking a key paragraph in which the authors fully explain the “birth” of the experimental number used LATER – only LATER – in the statistical approach.

The author pass time and words to describe certain statistical variables, but this interest is really secondary with respect to the method – fully reproduced for at least one donkey – which produce the numbers inserted in the experimental tables.

In this sense the 6 steps pf the experiments are really too similar each other and nothing is explained WHY the authors selected just these 6 steps. The recall a paper written in the past but this is not enough for the reviewer/reproducer of the experiment.

Please less statistical explanations and more accompany-learning dedicated to the first part of the work. The experimental one.

Moreover It is not enough to mention that the metereological data are from a X source: they are fundamental to be listed/graphed because is really not clear their  treatment.

Still not clear some variables – as the “moon phase at birth” is important to understand the relation between donkey behavior and i.e. a strong rain. When the strong rain occurred with respect to the experiments ? These time series are completely lacking in the “metereological paragraph”